

# DNN-based multi-output model for predicting soccer team tactics

Geon Ju Lee and Jason J. Jung

Department of Computer Engineering, Chung-Ang University, Seoul, Korea

## ABSTRACT

In modern sports, strategy and tactics are important in determining the game outcome. However, many coaches still base their game tactics on experience and intuition. The aim of this study is to predict tactics such as formations, game styles, and game outcome based on soccer dataset. In this paper, we propose to use Deep Neural Networks (DNN) based on Multi-Layer Perceptron (MLP) and feature engineering to predict the soccer tactics of teams. Previous works adopt simple machine learning techniques, such as Support Vector Machine (SVM) and decision tree, to analyze soccer dataset. However, these often have limitations in predicting tactics using soccer dataset. In this study, we use feature selection, clustering techniques for the segmented positions and Multi-Output model for Soccer (MOS) based on DNN, wide inputs and residual connections. Feature selection selects important features among features of soccer player dataset. Each position is segmented by applying clustering to the selected features. The segmented positions and game appearance dataset are used as training dataset for the proposed model. Our model predicts the core of soccer tactics: formation, game style and game outcome. And, we use wide inputs and embedding layers to learn sparse, specific rules of soccer dataset, and use residual connections to learn additional information. MLP layers help the model to generalize features of soccer dataset. Experimental results demonstrate the superiority of the proposed model, which obtain significant improvements comparing to baseline models.

## INTRODUCTION

The use of machine learning and data analysis in the field of sports is growing significantly. These are actively used for financial management of the club, medical management for injuries, and training of players. Meanwhile, research on players and game tactics is increasing (*Tianbiao & Andreas, 2016*; *Rein & Memmert, 2016*). Various machine learning and data analysis techniques are being applied to the field of soccer (*Yin & Kaynak, 2015*). The analysis of tactics and formations in soccer have not yet shown results. Soccer, unlike other sports, is dynamic, and the variables within a game are many. It is not easy to create and analyze events and features occurring in soccer games (*Park, Seo & Ko, 2016*).

The soccer market is rapidly increasing, with heavy capital investment. Studies for predicting tactics, formations, and soccer game outcome are being conducted. However, soccer coaches still rely on their own experience and intuition to determine game tactics

Corresponding author
Jason J. Jung, j2jung@gmail.com

| Table 1 Abbreviations of the positions. | |
|---|---|
| **Abbreviation** | **Positions** |
| FW | Forward |
| Wing | Winger |
| AMF | Attack Midfielder |
| CMF | Center Midfielder |
| DMF | Defense Midfielder |
| WB | Wing Back |
| CB | Center Back |
| GK | Goal Keeper |

and formations rather than data-based decision making (*Hill & Sotiriadou, 2016*). Even good coaches can make mistakes and bad decisions. The aim of this study is to predict the tactics, formations, and game outcome of a specific team, as well as to support the decision-making of the coach. In a previous study, we conducted to derive a synergistic relationship between players in a specific team using association rule mining. Players with high synergistic relevance were extracted and team formations and tactics were suggested (*Lee, Jung & Camacho, 2021*). In subsequent studies, the graph embedding technique was used. The team's formation and position information were converted into graph type and then into a vector by applying the graph embedding. The tactical formation of a team was analyzed by applying dimensionality reduction and clustering techniques to the embedding vector.

In this study, we predict the formation, game style, and game outcome of a given team. And we segment soccer dataset using machine learning techniques, such as feature selection and clustering, to effectively train the model. In existing studies, machine learning techniques, such as SVM and logistic regression, have mainly been used to predict tactics and game outcome in the field of soccer. However, these do not preprocess raw dataset and do not actively apply deep neural network-based models (*Rein & Memmert, 2016*). Tactics prediction of soccer in existing studies has the following limitations. First, they did not consider the types of soccer dataset. Soccer dataset has various features such as continuous, categorical, and binary type. Model training was not conducted effectively—input dataset was used without distinguishing the type of feature. Our model builds input layers according to the features. This makes it effective for learning representations of features. Also, existing studies did not segment soccer positions. In soccer, the same position is assigned different roles. For example, a striker (FW & Wing in Table 1) is assigned a specific role, such as a being a target man or shadow striker.

We segment soccer positions and use them to predict soccer tactics. The segmented positions make soccer tactics more diverse. Figure 1 shows the results obtained by applying feature engineering. The characteristics of each segmented position are represented by 0 and 1, where 0 is an offensive position and 1 is a defensive position. The criterion for dividing each position into 0 and 1 is based on the characteristics of the detailed role. For example, a target man is characterized by a high number of headers and a high number

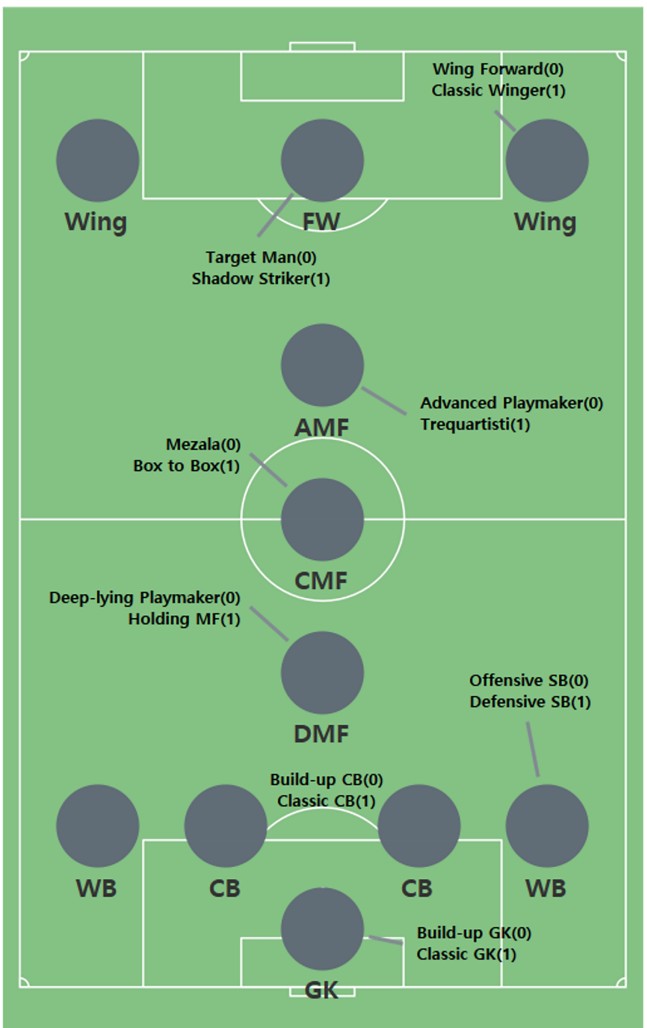

**Figure 1 Segmented positions by clustering.**

of shots. On the other hand, the Shadow Striker has high dribbling and passing counts. The roles are matched to each cluster by comparing the characteristics of the cluster with the characteristics of the detailed roles. The reason for expressing 0 and 1 in this study is to make it easier for readers to understand. Therefore, a detailed role of 0 represents an offensive player, and a detailed role of 1 represents a defensive player.

We use deep neural networks based on Multi-Layer Perceptron (MLP), wide inputs, and residual connections. Input layers is divided into continuous, categorical and binary features and widely applied to embedding and projection layers of the model. It is concatenated (*Cheng et al., 2016*) and utilizes a residual connection (*Lim et al., 2021*). Soccer tactics are determined by considering not only the formation but also factors such as segmented positions, game style, and game outcome. We propose a Multi-Output model for Soccer (MOS) that predicts the formation, game style, and game outcome. The most important core of soccer tactics is formation, segmented positions, and game style. To prepare for the game, the coach studies about soccer tactics such as formation, game style,

and segmented position (*Bradley et al., 2011*). Formation refers to a frame of soccer tactics such as 4-4-2, and segmented position refers to the role assigned to each player. The game style, whether offensive or defensive, reflects the overall direction of the game. Thus, the soccer coach can comprehensively consider the appropriate tactics.

The main contributions of the paper include:

1. We segment positions by applying machine learning techniques and present a method for effective training with small-sized dataset.
2. We propose a deep learning-based soccer analysis to predict soccer tactics, which has not been applied to the soccer.
3. We propose a model that can be applied to a framework to predict the soccer tactics.

The remaining section of this paper are as follows: "Related Work" introduces related studies. "Feature Engineering for Position Segmentation" describes the feature engineering method. "Multi-Output Model for Soccer Tactics" presents our model. "Experimental Results" focuses on the experimental results. Finally, "Conclusion and Future Works" concludes this study and presents future works.

## RELATED WORK

This work was inspired by previous studies. First, some studies have provided ideas for data modeling and segmentation inspired the feature selection technique using the Boruta algorithm to predict the audience demand for Major League Soccer (*Cheng et al., 2016*). The Boruta algorithm is used to extract important features from the stadium audience data to predict the the National Hockey League audience demand (*King, Rice & Vaughan, 2018*). Similarly, the outcome of the National Basketball Association was predicted using the Boruta algorithm to extract important features from player dataset (*Jain & Kaur, 2017*). A different study provided the theoretical explanation of the Boruta algorithm as well as the method of selecting important features (*Kursa & Rudnicki, 2010*).

In this study, clustering is applied using the important extracted features. In a study that presented the method of identifying the positions and formations of soccer players, a clustering technique was used to identify the positions and formations of players using the players' location information (*Bialkowski et al., 2016*). This concept, involving the application of clustering to the features of players, is also referenced into our study. In *Narizuka & Yamazaki (2018)*, the formation pattern is analyzed using the hierarchical clustering technique. Clustering is used to verify predicted soccer match results, and pass patterns are analyzed by clustering spatiotemporal soccer dataset (*Smeeton, 2003*). Among the clustering techniques, K-means is used in our study. Therefore, the study provides a theoretical explanation of K-means and a method for analysis.

In a study analyzing the difference in athletic performance between soccer players and basketball players, logistic regression was used to analyze the athlete exercise data. The study analyzed the athletic performance characteristics of soccer players and basketball players (*Chalitsios et al., 2019*). In a study analyzing events within a soccer game, the possibility of creating a goal event was analyzed. A study was conducted to determine the

likelihood of scoring a goal by analyzing features such as corner kicks, free kicks, counter attacks, defenders' reception, and play speed. The study quantified the effectiveness and strategy of each team (*Lucey et al., 2014*). The game outcome of each team was predicted using soccer feature data by applying machine learning techniques, such as support vector machine (SVM), decision tree, and logistic regression (*Cho, Yoon & Lee, 2018*). With a focus on databases in research predicting future match outcomes, a project called Open International Soccer Database was launched to provide resources to soccer analysts and benchmarks for machine learning methods (*Dubitzky et al., 2019*). The ranking of soccer players was predicted by selecting the model with the best performance among models such as SVM, MLP, Logistic, and Nave Bayes using ensemble learning (*Maanijou & Mirroshandel, 2019*).

Models such as MLP and RNN were applied to compare predictive performance using soccer players' game appearance dataset (*Strnad, Nerat & Kohek, 2017*). In a study that analyzes formation, the location of the players in each formation was specified by analyzing features such as the position and amount of activity of the players in each game (*Memmert et al., 2019*). The training model for our study was inspired by Wide and Deep Learning (WDL), which proposes wide linear model and deep neural network to combine the benefits of memorization and generalization for recommender systems (*Cheng et al., 2016*). WDL learns according to the type of input features and proposes a deep neural network based on feed forward networks. In our study, soccer data consists of several types of features and produces several outputs; thus, we refer to the architecture of WDL. Our model uses residual connections. We refer to Gated Residual Network (GRN) architecture and add a skip connection. Skip connection is a method of effectively learning the representation of data (*Lim et al., 2021*; *Cheng et al., 2016*).

In studies that visualize soccer formations, real soccer formations such as 4-3-2-1 are visualized (*Wu et al., 2018*). A study using video images of soccer visualized analyses such as team tactics and player evaluation (*Sheng et al., 2020*). In a study based on pass data from soccer games, the changing patterns of tactics were visualized by comprehensively analyzing pass data, including the related dynamics (*Xie et al., 2020*).

# FEATURE ENGINEERING FOR POSITION SEGMENTATION

Figure 2 shows the overview for tactic prediction. It mainly consists of 4 steps, which are *(i)* data collection, *(ii)* feature engineering, *(iii)* model training, and *(iv)* predictive analytics.

## Data collection

During the data collection, we focus on two kinds of datasets, which are (i) soccer player (in Table 2) and (ii) game appearance (in Table 3). Table 2 shows all possible features (*e.g.*, shooting, dribbling, and scoring) of the soccer player dataset. The soccer player dataset contains 68 features in total, and their performance (average values from all the season). The data provided for this experiment include 11 seasons and larger than the previous work (*Lee, Jung & Camacho, 2021*) which only 5 seasons were considered. The size of data

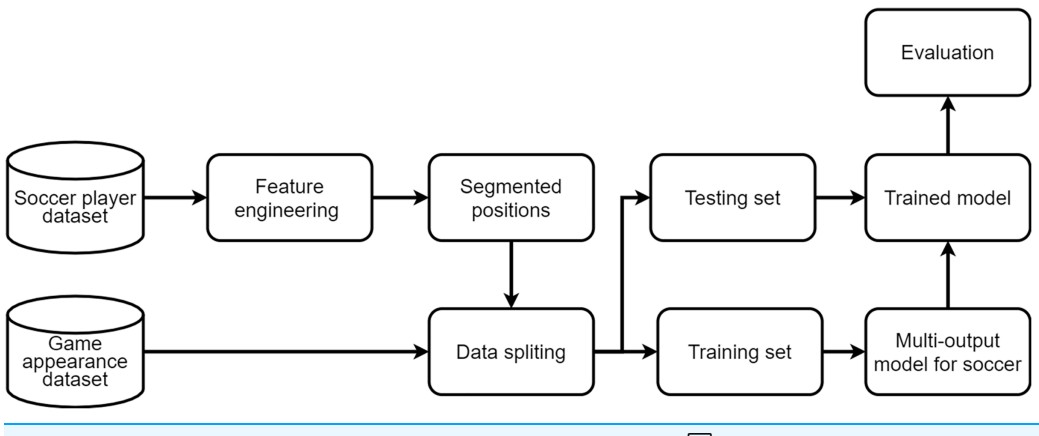

**Figure 2 Overview of the model training.**

**Table 2 Feature description of the soccer player dataset (*Lee, Jung & Camacho, 2021*).**

| Feature name | Feature description | Feature name | Feature description |
|---|---|---|---|
| Apps | Number of appearances | Mins | Appearances time |
| S_Total | Number of shots | S_OutOfBox | Number of shots outside the penalty zone |
| S_SixYardBox | Number of shots in the 6 yard box | S_PenaltyArea | Number of shots in the penalty zone |
| S_OpenPlay | Number of shots in open play situations | S_Counter | Number of shots in counterattack situation |
| S_SetPiece | Number of shots in a set piece situation | PenaltyTaken | Number of penalty kicks obtained |
| S_OffTarget | Number of invalid shots | S_OnPost | Number of shots hit by the goal |
| S_OnTarget | Number of effective shots | Blocked | Number of blocked shots |
| G_Total | Number of points | G_SixYardBox | Number of goals scored in the 6 yard box |
| G_PenaltyArea | Number of goals scored in penalty zone | G_OutOfBox | Number of goals scored outside the penalty zone |
| G_OpenPlay | Number of goals scored in open play situations | G_Counter | Number of goals scored in counterattack |
| G_PenaltyScored | Number of penalty kicks scored | G_Own | Number of own goals |
| G_Normal | Number of goals scored in normal situations | T_Dribbles | Number of dribble attempts |
| Successful | Number of dribble successes | Unsuccessful | Number of dribble failures |
| Touch Miss | Number of touch failures | Dispossessed | Number of lost possession of the ball |
| D_Total | Number of air contention | Won | Air Competition Wins |
| Lost | Air contention defeats | P_Total | Total number of passes |
| AccLB | Number of long pass successes | InAccLB | Number of long pass failures |
| AccSP | Number of short pass successes | InAccSP | Number of short pass failures |
| AccCR | Cross success number | InAccCR | Number of cross failures |
| InAccCrn | Number of successful corner kicks | InAccCrn | Number of corner kick failures |
| AccFrK | Number of free kicks successful | InFrK | Number of free kick failures |
| K_Total | Number of key passes | K_Long | Number of long key passes |
| K_Short | Number of short key passes | K_Cross | Number of cross key passes |
| K_Throughball | Number of Through Key Passes | A_Total | Number of assists |
| A_Cross | Number of cross assists | A_Corner | Number of corner kick assists |
| A_Throughball | Number of through-pass assists | A_Freekick | Number of free kick assists |
| T_Total | Number of tackle attempts | DribbledPast | Number of breakthroughs |
| Tackle | Number of tackle successes | Interception | Number of interceptions |

| Feature name | Feature description | Feature name | Feature description |
|---|---|---|---|
| Fouled | Number of being fouled | Fouls | Number of fouls |
| Offside | Number of offsides | Clearance | Number of cleared balls |
| ShotsBlocked | Number of blocking shots | CrossesBlocked | Number of Cross blocks |
| PassesBlocked | Number of passes blocked | Save_Total | Number of savings |
| Save_SixYardBox | Number of shooting saves outside the 6-yard box | PenaltyArea | Penalty Zone Shooting Savings |
| Save_OutOfBox | Number of savings by shooting outside the penalty zone | Rating | Average rating |

**Table 3** Features of the game appearance dataset.

| Binary features | Continuous features | Categorical features |
|---|---|---|
| FW0 FW0B FW1 | Ball Possession | Match order |
| AMF0 AMF0B AMF1 AMF1B | | Season |
| Wing0 Wing0B Wing1 Wing1B | | Opposing team |
| CMF0 CMF0B CMF0C CMF1 CMF1B | | Opp level |
| DMF0 DMF1 DMF1B | | |
| WB0 WB0B WB1 WB1B | | |
| CB0 CB0B CB1 CB1B | | |
| GK0 GK1 | | |

leads to the better performance of the predictions model which is the difference between this study and the previous one (*Lee, Jung & Camacho, 2021*).

For example, in the English Premier League, there are two FW players Harry Kane and Roberto Firmino. Harry Kane has shown higher performance of T_Dribbles (Number of dribble attempts) and S_Total (Number of shots) than Roberto Firmino has. Reversely, Roberto Firmino has shown higher with T_Total (Number of tackle attempts) and A_Cross (Number of cross assists) than Roberto Firmino has. We can understand a pair of features T_Dribbles and S_Total are significantly associated with each other, and these features are regarded as offensive performance. Also, T_Total and A_Cross are associated with each other as defensive performance.

Table 3 presents features of the game appearance dataset. The segmented positions such as FW0B, AMF1B, and CMF0B contain various information. For example, FW0B indicates that the position is FW, and the segmented position is 0 (Target Man). The segmented positions is shown in Fig. 1; B represents a specific player. Because several players in the same position can participate in a game, alphabet letters are added to distinguish them. Match order indicates the round of each game, and Opposing team indicates the opposing team for each game. Opp level indicates the level of the opposing team. Levels are divided based on the ranking of each season. Season represents the season of each game, and Ball Possession represents the ball possession in each game. For example, if Target Man (TM) Harry Kane and Shadow Striker (SS) Roberto Firmino play together in a game, FW0 and FW1 are included in the dataset.

## Feature engineering

In this study, we use feature engineering to generate a training dataset to be input to the proposed model, and feature engineering consists of feature selection and clustering. Feature selection is a machine learning technique that selects specific features from a large number of features. Inputting many features into the model results in over-fitting. To reduce the model complexity, the independent features that have greater influence on the dependent feature are selected. It is used to extract important features from high-dimensional data. The Boruta algorithm is a feature selection technique often used in the field of sports.

In previous work, implemented using the Boruta algorithm, data normalization was not taken in account. However, this study improved the model performance through data normalization. For example, G_Normal range from 0 to 0.8, while AccLB ranges from 0 to 50. The normalization utilizes process so that data can be obtained with the consistent scale attributes. It is the Min-Max with a value between 0 and 1 using 1. The formula for normalization with Min-Max technique is as follows:

$$x_{new} = \frac{x - x_{min}}{x_{max} - x_{min}} \tag{1}$$

where $x_{new}$ is the new value of data, and $x$ means old data. $x_{max}$ and $x_{min}$ are the maximum and the minimum value.

Algorithm 1 is a pseudo-code that describes the execution of the Boruta algorithm. In the pseudo-code, features such as shooting, dribbling, and scoring are defined as $s_i \in S$. Position is defined as $f_j \in F$. An important feature of a particular position is $S_{ij}$. The Boruta algorithm utilizes the *Z-score* of a random forest and is characterized by high diversity and high prediction accuracy. It achieves excellent stability and prediction accuracy by selecting important features based on various prediction results. The Boruta algorithm creates a new column, shadow features, by copying all features. The shadow features are randomly shuffled. A random forest is applied to the shadow features, and Max Z-score among Shadow Features (*MSZA*), which is the largest value among the derived *Z* values, is calculated. The *Z* value is derived by running a random forest on the existing features. If the *Z* value extracted from existing features is larger than *MSZA*, a specific feature is selected as an important feature.

Table 4 shows a part of the soccer player dataset for FW. For example, if Boruta algorithm is applied to the feature values of `Harry Kane`, `Roberto Firmino`, and `Jamie Vardy`, it produces *ShadowFeature* exactly like S_Total, S_OutOfBox, S_SixYardBox, S_PenaltyArea, and S_OpenPlay. It calculates *Z–Score* of each feature by applying the random forest. The feature with the highest *Z–Score* in *ShadowFeature* is *MZSA*. If *MZSA* is 15 and *Z–Score* of S_Total, S_OutOfBox, S_SixYardBox, S_PenaltyArea, and S_OpenPlay are 20, 16, 10, 7, and 18, respectively. Then, S_Total, S_OutOfBox, and S_OpenPlay, which have higher *Z–Score* than *MZSA*, are adopted as important features. In this study, we run the random forest 100 times in total.

---

**Algorithm 1  Boruta algorithm for feature selection.**

**Require:** *OriginalData* - Dataset(Position); *Runs* - the number of random forest runs.

**Ensure:** *ImportantSet* - Important record for a position

    *ConfirmedSet* = ø

    **for** *eachRuns* **do**

        *RecordData* ← *OriginalData*

        *ShadowFeature* ← permute(*RecordData*)

        *OriginalZ − Score* ← randomforest(*RecordData*)

        *Z − Score* ← randomforest(*ShadowFeatures*)

        *MZSA* ← *max(Z − Score)*

        **for** $F_j \in RecordData$ **do**

            **for** $s_i \in S$ **do**

                **if** $OriginalZ − Score(s_i) > MZSA$ **then**

                    *S*.append($s_i$)

                    *ConfirmedSet* ← $S_{ij}$

                **end**

            **end**

        **end**

    **end**

    **return** *ImportantSet* ← *ConfirmetSet*

---

**Table 4  Soccer player dataset for FW.**

| Features | Harry Kane | Roberto Firmino | Jamie Vardy |
|---|---|---|---|
| S_Total | 3.6 | 2.9 | 4.1 |
| S_OutOfBox | 0.7 | 0.8 | 0.9 |
| S_SixYardBox | 0.4 | 0.5 | 0.2 |
| S_PenaltyArea | 1.4 | 2.1 | 1.7 |
| S_OpenPlay | 2.3 | 2.8 | 1.9 |
| Rating | 7.7 | 6.8 | 7.3 |

After extracting important features from the soccer player dataset using the Boruta algorithm, clustering is performed based on the important features. In this study, we use K-means among clustering methods. K-means calculates the distance of each vector based on a centroid. The centroid moves to determine the minimal location from the vector, at which point clustering is completed. In clustering, we use the Elbow method. It detects the change in Sum of Squared Error according to the number of K. We determine the number of K at which the sum of squared errors becomes constant. The purpose of Eq. (2)

 

**Table 5 Segmented positions for FW.**

| Players | Harry Kane | Roberto Firmino | Jamie Vardy |
|---|---|---|---|
| Label (Segmented position) | 0 | 1 | 0 |

is to find a cluster where the sum of squared distances of $\mu_k$ and $S_{ijn}$ are minimized. K-means is given as

$$\arg\min_t \sum_{i=1}^{k} \sum_{S_{ijn} \in T_k} \| S_{ijn} - \mu_k \|^2 \tag{2}$$

where $S_{ijn} \in S$ means a specific player $n$ of a specific position $j$ and important feature $i$, and $\mu_k$ denotes the centroid of $T_k$. The objective of the K-means algorithm is to assign clusters to sets $T = \{t_1, t_2, \ldots, t_k\}$.

Table 5 shows a part of segmented positions for FW. S_Total, S_OutOfBox, and S_OpenPlay are important features extracted from the Boruta algorithm. We cluster `Harry Kane`, `Roberto Firmino`, and `Jamie Vardy` based on the extracted important features. `Harry Kane` and `Jamie Vardy` have higher S_Total and lower S_OpenPlay than `Roberto Firmino`. In K-means, clustering is calculated based on the difference in each feature value. `Harry Kane` and `Jamie Vardy` are classified into the same cluster since they have similar feature values. Each player is given a label of 0 or 1 based on its cluster. This indicates the segmented position of each player.

## MULTI-OUTPUT MODEL FOR SOCCER TACTICS

In this study, our model predicts multiple outputs. A soccer coach considers various factors when determining team's tactics. For example, formation, game style, and detailed roles of each player are comprehensively considered. Our model predicts the most important factors in soccer tactics: formation and game style. 4-3-2-1 is an example of a formation, and game style is divided into offensive or defensive style. Finally, we predict the outcome of each game.

We build the model with reference to models from WDL and GRN (*Lim et al., 2021*; *Cheng et al., 2016*). Our dataset includes binary, continuous, and categorical features. When we learn multiple types of data in one layer, the performance decreases. We took ideas from WDL and GRN models to improve performance and generalize the model. WDL combines a non-linear deep model specialized for generalization. In the deep model, interactions between features are expressed in a non-linear space. Therefore, it has strong characteristics in generalization. Skip connection is used in GRN model. Skip connection allows each layer to learn a small amount of additional information. The amount of information each layer has to learn is reduced. By linking the output of the previous layer to a specific layer, only the information that needs to be learned additionally is mapped. Our model has the advantage of learning faster by using skip connection and can reduce the size of the error. The model structure is shown in Fig. 3. The key difference between our model and WDL, GRN is that we add skip connections to learn features for soccer dataset, and we build a model of multiple outputs rather than one output. Further

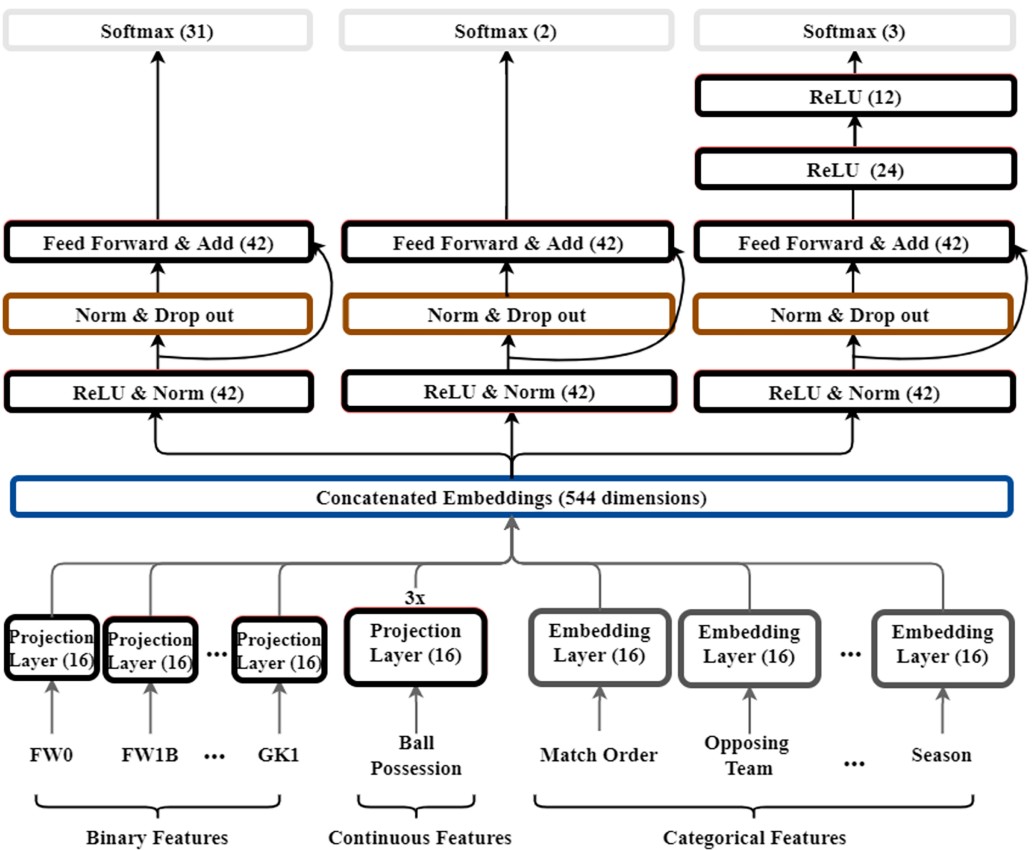

**Figure 3 Structure of multi-output model for soccer.** The numbers in this figure mean the dimensions of each layer. Also, the color of each layer is indicated to distinguish the role of the layer. Red color indicates the feed-forward layer as dense layer, black indicates the embedding layer, green indicates batch normalization and drop-out, and yellow indicates the output layer.

feed forward networks of each output layer are added. A projection layer is added to effectively learn the representation of each feature according to the type of inputs. In the following parts, we introduce, in a bottom–up manner, the key components of our model: embedding layer, projection layer, and MLP.

## Embedding layer and projection layer

In this study, input features have a total of three data types: binary, continuous, and categorical. Features such as ball possession rate are continuous and segmented positions such as FW0 and CMF1B are binary features. Categorical features include the order of the match, opposing team, and level of the opposing team. They are often encoded as one-hot vectors and sparse vectors but, this leads to high-dimensional feature spaces. To reduce the feature spaces, we add embedding layers to transform these categorical features into dense vectors as embedding vectors

$$X_{embed,i} = W_{embed,i}X_i \qquad (3)$$

where $X_{embed,i}$ is the embedding vector, $X_i$ is the input in the categorical features, $W_{embed,i} \in \mathbb{R}^{n_e \times n_v}$ is the embedding matrix, $n_e$ is the embedding size, and $n_v$ is vocabulary size. Continuous features and binary features are fed to the projection layer and do not pass through the embedding layers. We add projection layers to transform continuous and binary features into dense vectors

$$X_{projection,i} = W_{projection,i} X_i \tag{4}$$

where $X_{projection,i}$ is the dense vector from projection layers, $X_i$ is the binary and continuous input in the binary and continuous features, $W_{projection,i} \in \mathbb{R}^{n_u \times n_i}$ is the projection matrix, $n_u$ is the unit size of projection layer, and $n_i$ is input size. However, the number of projection layers that binary features and continuous features pass through is different. Binary features pass through one projection layer, whereas continuous features pass through three projection layers to deep learn the representation of the features. The vectors passing through the embedding layer and the projection layer are merged into one vector. We stack the dense vectors along with the embedding vectors into one vector

$$X_0 = Concat(X_{embed,1}, \ldots, X_{embed,k}, X_{projection,1}, \ldots, X_{projection,k}) \tag{5}$$

where $X_0$ is the concatenated vector.

## MLP layers and loss function

By concatenating the dense vectors and embedding vectors into one vector, we use fully connected layers to learn the relations among the features. Our model yields three outputs. Each output layer goes through fully connected layers. We also apply a residual connection. Residual connections add output from previous layers to help represent learning. It is mainly used in the image processing field, but it is also used for learning structured data. Therefore, our model trains the model by applying a residual connection. This can contribute to our model training faster and reducing the size of error. The MLP layers are a feed-forward network, with each layer having the following formula

$$a^{(l+1)} = f(W^{(l)} X_0 + b^{(l)}) \tag{6}$$

where $l$ is the layer number and $f$ is ReLU (the REctified Linear Unit) function. $X_0$, $b^{(l)}$, and $W^{(l)}$ are the concatenated vector, bias, and model weights at $l$-th layer, respectively. To avoid overfitting and learn meaningful features, we add *BatchNorm*, *dropout*, and ReLU. The overall output of the MLP layers is as

$$F_1 = BatchNorm(Dropout(ReLU(W^{(l)} X_0 + b^{(l)}))) \tag{7}$$

where $W^{(l)}$, $b^{(l)}$ are the learnable parameters, and *BatchNorm* is the standard normalization layer. *Dropout* is the layer for avoiding overfitting of the model. After passing through the MPL layers, we make a residual connection to the output of the previous layer. The output of the residual connection is as

$$F_2 = F_1 + ReLU(BatchNorm(Dropout(ReLU(W^{(l)}X_0 + b^{(l)})))W^{(l+1)} + b^{(l+1)}) \qquad (8)$$

where $W^{(l)}$ and $W^{(l+1)}$ are the $l$-th and $(l + 1)$-th feed-forward layer, respectively; and $F_1$ is the output of the previous feed-forward. Therefore, the output of the previous layer and the output of the next layer are added. The last layer is a Softmax layer. It is expressed as a probability value of the output to be predicted. To train the model, we use the cross-entropy loss with a regularization term

$$loss = -\frac{1}{N}\sum_{i=1}^{N} y_i \log(p_i) + (1 - y_i)\log(1 - p_i) + \lambda \sum_l \|W^{(l)}\|^2 \qquad (9)$$

where $p_i$ are the probabilities computed from Softmax layer, $y_i$ represents the labels, $N$ is the total of inputs, and $\lambda$ is the $L_2$ regularization parameter. We use a cross-entropy loss function. Unlike formation and game style prediction, game outcome prediction requires more feed-forward networks. Therefore, two feed-forward layers are added to the game outcome layer.

In conclusion, our model trains according to binary, categorical, and continuous types. The dataset goes through a projection layer, an embedding layer, and feed-forward networks. Our model outputs formations (such as 4-4-2 and 4-5-1), game styles (such as offensive and defensive), and game outcome (such as win, draw, and loss).

## EXPERIMENTAL RESULTS

In this section, we present the experimental settings and results. And we evaluate the performance of our model on the soccer dataset.

### Experimental settings

We collected two datasets from the website (http://whoscored.com). The first dataset is the soccer player dataset, as described in Table 2. The second dataset is the game appearance dataset in Table 3. We combine two datasets into a training dataset. We collect Tottenham Hotspur's game appearance dataset for 11 seasons (2010/2011–2020/2021). Our model trains a dataset consisting of 380 games and tests the model with a dataset consisting of 38 games. The data from 2010/2011 to the 2019/2020 seasons is used as training data, while the data from 2020/2021 season is used as test data.

We trained the model using the training dataset for 100 epochs. We used the Adam optimizer, a dropout rate of $P_{drop} = 0.1$, and a learning rate of 0.01. We applied dropout to the output of each sublayer and $L_2$ regularization (weight decay) of 0.001 on a fully connected layer. The model parameter details are presented in Table 6.

### Results on feature engineering

We applied the Boruta algorithm to extract important features from the soccer player dataset. Random forest iteration was performed 100 times. Table 7 shows the result. For FW, many offensive features such as S_Total, S_SixYardBox, and T_Dribbles were selected, and for AMF, features related to passing were adopted. For CMF and DMF, Offensive, Passing, and Defensive features were uniformly selected, and for Wing, all

**Table 6 Configuration of our model.**

| Configuration of our model | | | |
|---|---|---|---|
| Embedding size | 16 | batch size | 24 |
| Dropout | 0.2 | #epochs | 100 |
| $L_2$ regularization | 0.001 | optimizer | *Adam* |
| Learning rate | 0.01 | | |

**Table 7 Features selected from Boruta algorithm (*Lee, Jung & Camacho, 2021*).**

| Position | | Category | Selected features |
|---|---|---|---|
| FW | FW | Offensive | S_Total, S_SixYardBox, S_PenaltyArea, S_OpenPlay, S_SetPiece, S_OffTarget, S_OnTarget, G_Total, G_SixYardBox, G_PenaltyArea, G_OutOfBox, G_OpenPlay, G_SetPiece, G_Normal, T_Dribbles, Unsuccessful, Successful, D_Total, Won |
| | | Passing | P_Total, AccLB, AccSP, InAccSP, K_Total, K_Short |
| | | Defensive | ShotsBlocked |
| MF | AMF | Offensive | S_Total, S_OutOfBox, S_PenaltyArea, S_OpenPlay, Blocked, G_Total, G_PenaltyArea, G_OutOfBox, G_OpenPlay, G_Normal, Successful, D_Total |
| | | Passing | P_Total, AccLB, AccSP, InAccSP, AccCr, InAccCr, InAccCrn, AccFrk, K_Total, K_Long, K_Short, K_Cross, K_Corner, K_Throughball, A_Total, A_Cross, A_Corner |
| | | Defensive | T_Total |
| | CMF | Offensive | S_Total, S_OutOfBox, S_SixYardBox, S_PenaltyArea, S_OpenPlay, S_OffTarget, S_OnTarget, Blocked, G_Total, G_PenaltyArea, G_OutOfBox, G_OpenPlay, G_Normal, Unsuccessful, Successful, T_Dribbles, D_Total, Won, Lost |
| | | Passing | P_Total, AccSP, InAccSP, ACCCr,AccCrn, K_Total, K_Short, K_Throughball, A_Total, A_Throughball |
| | | Defensive | T_Total, Interception, PassesBlocked, Tackle |
| | DMF | Offensive | S_Total, S_OutOfBox, S_SixYardBox, S_PenaltyArea, PenaltyTaken, S_OffTarget, S_OnTarget, G_Total, G_PenaltyArea, Successful, Unsuccessful, Dispossessed, D_Total, Won |
| | | Passing | P_Total, AccSP, AccFrK, K_Total, K_Long, K_Short, K_Throughball, A_Total |
| | | Defensive | T_Total, Interception, Tackle |
| | Wing | Offensive | S_Total, S_OutOfBox, S_SixYardBox, S_PenaltyArea, S_OpenPlay, S_OnTarget, Blocked, G_Total, G_PenaltyArea, G_OutOfBox, G_OpenPlay, G_normal, Unsuccessful, Successful, T_Dribbles, Won |
| | | Passing | P_Total, AccLB, AccSP, InAccSP, AccCr, InAccCr, AccCrn, InAccCrn, InAccFrk, K_Total, K_Long, K_Short, K_Cross, K_Corner, K_Throughball, K_Freekick, A_Cross, A_Total |
| | | Defensive | T_Total, Interception, Fouled, ShotsBlocked, Tackle |
| DF | WB | Offensive | S_Total, S_SixYardBox, S_PenaltyArea, S_OpenPlay, S_SetPiece, S_OnTarget, Blocked, G_Total, G_OutOfBox, G_SetPiece, G_Normal, Successful, T_Drribles, D_Total, Won, Lost |
| | | Passing | P_Total, AccSP, AccCr, InAccCr, K_Total, K_Long, K_Short, K_Cross, A_Cross, A_Total |
| | | Defensive | T_Total, DrribledPast, Tackle, Interception, Fouled, ShotsBlocked, PassesBlocked |
| | CB | Offensive | S_Total, S_OutOfBox, S_PenaltyArea, S_OpenPlay, S_SetPiece, S_OffTarget, S_OnTarget, G_Total, G_SetPiece, G_Own, G_Normal, Successful, T_Dribbles, D_Total, Won, Lost |
| | | Passing | P_Total, AccSP, K_Total, K_Short |
| | | Defensive | T_Total, Interception, Fouled, Clearance, Tackle |
| GK | GK | Passing | InAccLB, AccSP, AccFrK |
| | | Defensive | Clearance |
| | | Saving | Save_Total, Save_PenaltyArea, Save_OutOfBox |

**Table 8 The numbers of the clusters by elbow method. We have empirically decided the numbers of these clusters.**

| Position (# cluster) | Position (# cluster) | Position (# cluster) | Position (# cluster) |
| --- | --- | --- | --- |
| FW (2) | AMF (2) | Wing (2) | CMF (2) |
| DMF (2) | SB (2) | CB (2) | GK (2) |

features were comprehensively adopted. Although WB is DF, more offensive features are selected than CB, and for CB, features related to defense were selected. GK did not have a large range of activity, so it was mainly selected for defensive features.

Table 8 shows the number of clusters for each position. Two clusters are generated for each position, and each cluster represents a segmented position. The visualization of the segmented position is shown in Fig. 1. In FW, the dribbling ability of Target Man (TM) is poor, whereas their ability to compete for aerial balls is excellent; TM has a large build and lacks agility but can provide chances to teammates by competing with opposing defenders in the air based on their physical characteristics. On the other hand, Shadow Striker (SS) has a lot of dribbling attempts.

## Model evaluation and discussion

In this section, we evaluate four models. The first is a simple feed-forward network, a baseline model that predicts three outputs from one input. The difference between it and our model is that the input layer according to the data type is not separately configured. The baseline model does not use residual connection and deep dense layers. There is no batch normalization, dropout, and $L_2$ regularization. To show the effectiveness of our model, we compare it with two models: WDL (*Cheng et al., 2016*) and GRN (*Lim et al., 2021*). We compare the proposed model with the three baseline models. Our model is evaluated in two cases: with or without feature engineering. Feature engineering is an important aspect of this study. Therefore, we evaluate our model performance according to feature engineering. The evaluation metric is accuracy. Also, cross validation methods include Hold-out, K-fold, Stratified K-fold, and Repeated Random in the experiment. Cross validation is a method that cross-changes training data and test data. All data can be used for training, and underfitting from insufficient data can be prevented. It provides more robust model performance. The dataset used in the experiment is not large. Therefore, there is a problem that the performance is not stable. To solve this problem, four cross validation methods are applied, and the method with the most stable performance is adopted. Hold-out divides training and test data at a certain ratio and uses it for learning. K-fold divides the entire dataset into K folds and performs training. This is the most common cross validation method. Stratified k-fold is a method to improve performance evaluation when data is biased. Repeated random validation splits the dataset randomly into training and test. All model evaluations are tested with the same hyperparameters, 100 epochs, and a learning rate is of 0.01. Performance metrics include accuracy, precision, and recall. Which are listed in Table 9.

**Table 9 The performance evaluation metrics.**

| Metrics | Description | Formula |
|---|---|---|
| Accuracy | This is the sum of TP and TN divided by the total sum of the population | $\dfrac{TP + TN}{TP + TN + FP + FN}$ |
| Precision | It is TP divided by the total sum of TP, FP | $\dfrac{TP}{TP + FP}$ |
| Recall | This is TP divided by the total sum of TP, FN | $\dfrac{TP}{TP + FN}$ |

**Table 10 Performance evaluation of model accuracy from cross-validation methods (baseline, WDL, GRN, our model without feature engineering, our model with feature engineering).**

| Method | Model | Formation Acc | Game style Acc | Game outcome Acc |
|---|---|---|---|---|
| Hold-out validation | Baseline | 74.35% | 72.4% | 42.5% |
| | WDL | 91.78 | 82.14 | 57.96 |
| | GRN | 93.54 | 83.65 | 56.64 |
| | Our model (Without F.E.) | 60.58 | 58.1 | 34.2 |
| | Our model (With F.E.) | 93.67 | 83.35 | 57.59 |
| 5-fold cross validation | Baseline | 73.58% | 69.57% | 43.51% |
| | WDL | 90.85 | 81.87 | 56.85 |
| | GRN | 94.84 | 83.62 | 57.78 |
| | Our model (Without F.E.) | 59.34 | 57.76 | 33.1 |
| | Our model (With F.E.) | 94.97 | 83.82 | 57.82 |
| Stratified 5-fold cross validation | Baseline | 71.64% | 70.26% | 42.41% |
| | WDL | 90.69 | 81.23 | 56.94 |
| | GRN | 93.11 | 82.49 | 55.81 |
| | Our model (Without F.E.) | 59.28 | 54.32 | 34.97 |
| | Our model (With F.E.) | 94.85 | 82.97 | 56.16 |
| Repeated random validation | Baseline | 70.86% | 68.38% | 43.76% |
| | WDL | 89.85 | 82.47 | 56.81 |
| | GRN | 93.94 | 82.57 | 56.21 |
| | Our model (Without F.E.) | 57.58 | 53.88 | 33.62 |
| | Our model (With F.E.) | 93.53 | 83.07 | 57.54 |

Table 10 shows the performance of the models according to the cross-validation method. The baseline model records modest accuracy in predicting formations and game styles. However, it shows a fairly low prediction accuracy for the game outcome. It shows the lowest performance in all cross validation methods. Second, the WDL model shows significantly higher accuracy than the baseline. In particular, the accuracy of the formation and game style has been greatly improved. The performance of the GRN model is significantly higher than the baseline. It shows slightly improved performance than WDL models. On the other hand, our model trained without feature engineering shows the lowest accuracy. Our model without feature engineering does not learn the representation of the data properly. In other words, it indicates that feature engineering of datasets can be

**Table 11 Performance evaluation of model precision from 5-fold cross-validation (baseline, WDL, GRN, our model without feature engineering, our model with feature engineering).**

| Method | Model | Formation precision | Game style precision | Game outcome precision |
|---|---|---|---|---|
| 5-fold cross validation | Baseline | 72.07% | 70.61% | 45.43% |
| | WDL | 91.81 | 82.97 | 56.07 |
| | GRN | 93.51 | 81.72 | 56.53 |
| | Our model (Without F.E.) | 56.75 | 53.76 | 35.60 |
| | Our model (With F.E.) | 93.84 | 83.35 | 56.26 |

**Table 12 Performance evaluation of model recall from 5-fold cross-validation (baseline, WDL, GRN, our model without feature engineering, our model with feature engineering).**

| Method | Model | Formation recall | Game style recall | Game outcome recall |
|---|---|---|---|---|
| 5-fold cross validation | Baseline | 70.09% | 66.37% | 42.93% |
| | WDL | 90.15 | 80.83 | 53.17 |
| | GRN | 90.58 | 80.94 | 54.33 |
| | Our model (Without F.E.) | 57.06 | 56.98 | 32.79 |
| | Our model (With F.E.) | 91.92 | 81.18 | 54.26 |

quite important in deep learning. Finally, our model trained with feature engineering shows the highest accuracy. However, the performance is not significantly different from other models. As a result, all models had the highest formation accuracy, followed by game style accuracy. However, the accuracy of the game outcome is relatively low. In this experiment, a total of 4 cross validation methods were applied, and the 5-fold cross validation method has slightly stable performance. Also, All baselines were trained through feature engineered datasets. To clarify the performance difference, we tested by training the feature-engineered dataset and non-feature-engineered dataset only in our model separately. In most models, it is confirmed that the prediction of the formation and game style is relatively high, but the prediction accuracy of the game outcome is low. The unprocessed dataset does not have segmented positions; thus, it is difficult for the model to learn. The performance of the model can be improved through feature engineering. By configuring the input features according to the type of data and using a residual connection, deep feed-forward networks, the performance of the model can be improved.

Tables 11 and 12 show precision and recall. Overall, it shows a similar trend to accuracy. However, precision and recall need to be grasped in detail according to each predicted class. Therefore, the prediction results of formation, game style, and game outcome should be looked at individually.

Figure 4 is the confusion matrix of formation predicted from our model. Overall, it shows good accuracy in all classes. It shows stable performance because there is little bias for formation in the dataset.

Figure 5 is the confusion matrix of game style predicted from our model. This experiment was conducted based on the game data of Tottenham Hotspur. Tottenham
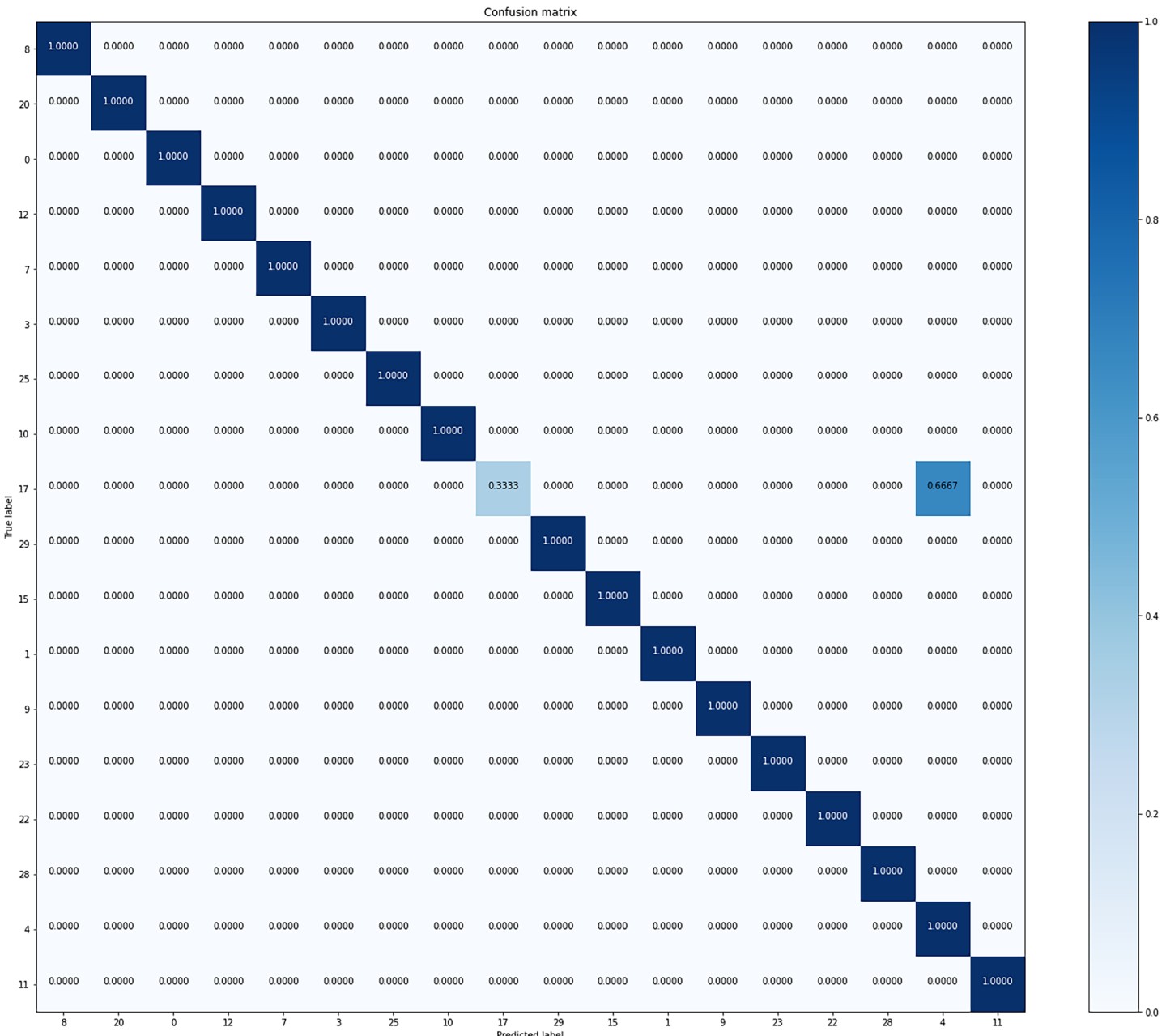

**Figure 4 Confusion matrix of our model from the best performing method (5-fold cross validation) for formation classification.**

Hotspur are a top-middle-class team in the English Premier League. Therefore, the possession of the ball in the game data is higher than that of other teams. The prediction for the offensive class is stable. On the other hand, the prediction performance for the defensive class is low. The defensive class is often mistakenly predicted as offensive.

Figure 6 is a confusion matrix of game outcomes predicted from our model. Overall, the prediction for the Draw class is good. Also, the rate of predicting lose and win as draw is

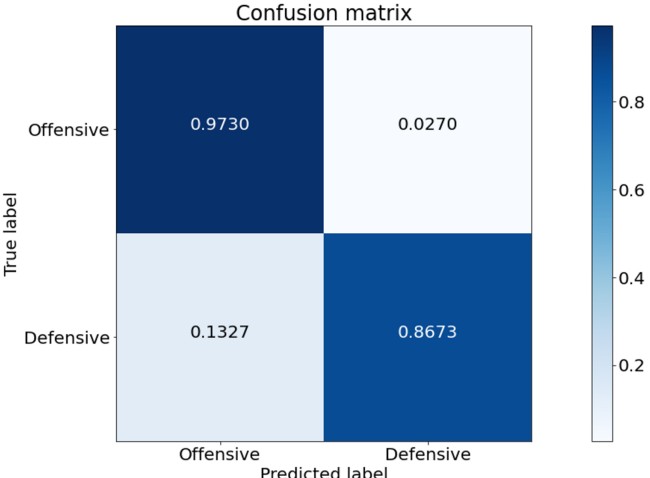

**Figure 5 Confusion matrix of our model from the best performing method (5-fold cross validation) for game style classification.**

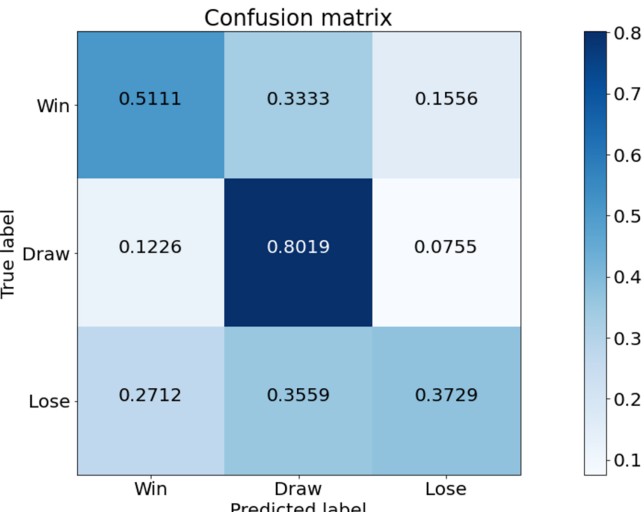

**Figure 6 Confusion matrix of our model from the best performing method (5-fold cross validation) for game outcome classification.**

quite high. This indicates that the draw class distribution is high in the data used in the experiment. The class with the worst predictive performance is lose. Tottenham Hotspur is a top-middle-class team, so there is less data on lose compared to win and draw. Therefore, the predictive performance of lose is poor.

## Results of multi-output model for soccer

To verify the performance of the proposed multi-output model for soccer, we have conducted feasibility testing with the predicted results. Thus, we have found out how tactical decision (such as formation, positions and game style) can be changed, according to features such as the level of opposing team. Three teams Manchester City (the top ranked team as difficult level), Everton (the 10th ranked team as middle level), and

**Table 13 Input dataset for model prediction.**

| Features | vs. Manchester City | vs. Everton | vs. Sheffield |
|---|---|---|---|
| FW0 | 1 | 1 | 1 |
| FW0B | 0 | 0 | 0 |
| FW1 | 0 | 0 | 0 |
| AMF0 | 0 | 0 | 0 |
| AMF0B | 0 | 0 | 0 |
| AMF1 | 1 | 0 | 1 |
| AMF1B | 0 | 0 | 0 |
| Wing0 | 1 | 1 | 1 |
| Wing0B | 1 | 0 | 0 |
| Wing1 | 0 | 1 | 1 |
| Wing1B | 0 | 0 | 0 |
| CMF0 | 0 | 1 | 1 |
| CMF0B | 0 | 0 | 1 |
| CMF0C | 0 | 0 | 0 |
| CMF1 | 0 | 1 | 0 |
| CMF1B | 0 | 0 | 0 |
| DMF0 | 1 | 1 | 0 |
| DMF1 | 1 | 1 | 1 |
| DMF1B | 0 | 0 | 0 |
| WB0 | 1 | 0 | 0 |
| WB0B | 0 | 0 | 0 |
| WB1 | 1 | 0 | 0 |
| WB1B | 0 | 0 | 0 |
| CB0 | 1 | 1 | 1 |
| CB0B | 1 | 1 | 1 |
| CB1 | 0 | 1 | 1 |
| CB1B | 0 | 0 | 0 |
| GK0 | 1 | 1 | 1 |
| GK1 | 0 | 0 | 0 |
| Opp level | High | Mid | Low |
| Match order | 9 round | 32 round | 18 round |
| Season | 2020–2021 | 2020–2021 | 2020–2021 |
| Opposing Team | Manchester City | Everton | Sheffield |
| Ball Possession | 66.1 | 47.2 | 58 |

Sheffield United (the 20th ranked team as easy level). Table 13 shows the input dataset for model prediction more in details.

Moreover, we have developed the visualization system (http://recsys.cau.ac.kr:8092/) for better understanding on the predicted results (the formation, game style, and game outcome).

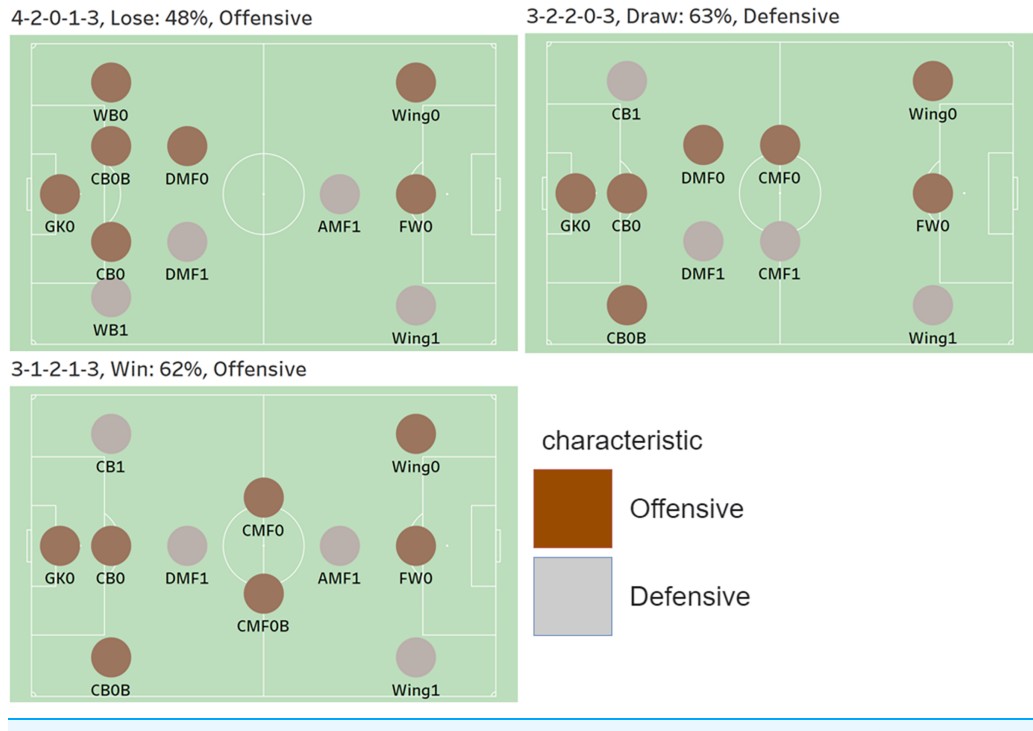

**Figure 7 Visualization against three cases, which are Manchester City (9 Round), Everton (32 Round), and Sheffield (18 Round).**

**Table 14 Prediction results against three cases, which are Manchester City (9 Round), Everton (32 Round), and Sheffield (18 Round).**

| Results | vs. Manchester City | vs. Everton | vs. Sheffield |
|---|---|---|---|
| Actual formation | 4-2-0-1-3 | 3-2-2-0-3 | 3-1-2-1-3 |
| Actual game style | Offensive | Defensive | Offensive |
| Actual game outcome | Lose | Win | Win |
| Predicted formation | 4-2-0-1-3 | 3-2-2-0-3 | 3-1-2-1-3 |
| Predicted game style | Offensive | Defensive | Offensive |
| Predicted game outcome | Lose | Draw | Win |

Figure 7 visualization against three cases, which are Manchester City (9 Round), Everton (32 Round), and Sheffield (18 Round). The first visualization is in the case of the 1st team, Manchester City. Each position is represented by a colored dot, and the color represents the characteristic of each position. For example, a position labeled 0 is an offensive position, and 1 is defensive. Labels 0 and 1 are the result of segmentation through feature engineering. Additionally, the predicted formation against Manchester City is 4-2-0-1-3, and the predicted game outcome is a loss. The probability of defeat appears to be 48%. Further, the game style is predicted to be offensive. The second is a visualization based on the predicted formation, game style, and game outcome in the case of Everton, the 10th placed team. The predicted formation against Everton is 3-2-2-0-3, and the game style is predicted to be defensive. Further, the game outcome has the highest

probability of a draw with 63%. The last shows the prediction results for the 20th place team, Sheffield United. The predicted formation is 3-1-2-1-3, and the game outcome has the highest probability of winning with a 62% chance. Moreover, the game style is predicted to be aggressive.

Table 14 shows the actual and predicted results. The predictions for the formation and game style have been matched with the actual ones, while the predicted game outcome against Everton has not.

## CONCLUSION AND FUTURE WORKS

We used feature selection and clustering techniques to segment positions. We configured the input layers according to the type of data and adjusted the depth of the layers according to the output. We constructed a wide and deep model for learning soccer dataset and used residual connections for effective learning. In this study, we successfully predicted soccer tactics using soccer dataset and suggested the best tactics for each game. We developed a methodology to predict the tactics of a game using a small amount of data. This study proposed a method for predicting tactics through segmentation of soccer positions and deep learning modeling on small-sized dataset in the field of soccer. It is possible to predict the tactics for a soccer game comprehensively through a model that generates multiple outputs instead of one output.

However, there are some shortcomings in this study. The first is that we could not utilize datasets from any team other than Tottenham Hotspur. In this study, an experiment was conducted using the game dataset of Tottenham Hotspur in the English Premier League. It is expected that more meaningful results would have been obtained if dataset from many teams were utilized. In addition, the prediction performance for game outcome was relatively low. Because there are so many variables in soccer, predicting game outcome is difficult compared to other sports. However, many studies are currently being conducted to predict the outcome of soccer games. Future research will put more effort into data collection and devise ways to improve the performance of match results prediction. We will develop a methodology that can predict and suggest various soccer tactics.

### Funding
This research was supported by the Chung-Ang University Graduate Research Scholarship in 2020. The funders had no role in study design, data collection and analysis, decision to publish, or preparation of the manuscript.

### Grant Disclosures
The following grant information was disclosed by the authors:
Chung-Ang University Graduate Research Scholarship.

## Competing Interests

Jason J. Jung is an Academic Editor for PeerJ.

## Author Contributions

- Geon Ju Lee conceived and designed the experiments, performed the experiments, analyzed the data, performed the computation work, prepared figures and/or tables, authored or reviewed drafts of the paper, and approved the final draft.
- Jason J. Jung performed the experiments, analyzed the data, performed the computation work, authored or reviewed drafts of the paper, and approved the final draft.

## Data Availability

The data and source code are available at GitHub: https://github.com/kecau/Multi-Output-model-for-Soccer.

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
