# Peer review of "DNN-based multi-output model for predicting soccer team tactics"

_PeerJ Computer Science, doi:10.7717/peerj-cs.853_

## Round 0.1 · original submission · Minor Revisions

Both reviewers agree the paper needs minor revisions. Please prepare a new version of the document according their suggestions, as well as a report explaining how you have addressed them.

Reviewer 1 ·

Basic reporting

> The authors proposed to use Deep Neural Networks and feature engineering to predict the soccer tactics of teams, such as formations, game styles, and game outcome based on soccer dataset.

> The authors stated that experimental results demonstrated significant improvements of the proposed model compared to baseline models.

> The manuscript is literature supported.

> A new English revision is encouraged, e.g:

Line 24: “the proposed model, which obtain” , missing “s”
Line 66: “We propose Multi-Output” , missing “a”
Line 67: “game outcome; The” , missing ".”
Line 76: “We propose deep learning” , missing “a”
Line 195: “determining a team’s tactics” , remove “a”

Experimental design

> Source code and data source were provided. Also, Python was used as programming language, which helps to reach a bigger audience.


> “Our model trains a dataset consisting of 380 games and tests the 233 model with a dataset consisting of 38 games.”

It seems that hold-out validation was applied, or a single cross-validation. However, it is well-known that there are other more robust validation methods, such as leave-one-out and k-fold cross-validation. Also, the raw data provided in https://github.com/Lee-Gunju/Multi-Output-Model-for-Soccer-based-on-DNN-and-Feature-Engineering-for-Predicting-Team-Tactics do not seem difficult to process.

Can the authors extend the experimental study? For example, by adding new datasets and using a different validation method.


> “Moreover, we have developed the visualization system”

Apologies, during the revision, the provided system was unavailable (http://recsys.cau.ac.kr:8092/). Please check the provided figures.

Validity of the findings

> After reviewing and testing the provided source code (https://github.com/kecau/Multi-Output-model-for-Soccer).

Please update how the data is loaded in Line 10, since xlrd has explicitly removed support for anything other than xls files. You can follow this link https://stackoverflow.com/questions/65254535/xlrd-biffh-xlrderror-excel-xlsx-file-not-supported.

> Source code is missing from the file “Multi-Output model for Soccer.py”. For example, when executing your script the following error is seen: “NameError: name 'train_dt' is not defined”.

> Why the 2nd output (game styles) used a softmax instead of a sigmoid activation?

> Regarding the game outcome performance, the results achieved are discrete (49.39% recall). However, the authors surpassed the baseline models. The authors should extend the experimental analysis in order to better support their proposal.

> Please include the individual sensitivity rates achieved for every outcome in the analysis in order to have a complete picture of your proposal, e.g. win, draw, loss, offensive, defensive.

> The conclusion should be stated after performing more experiments with new datasets and state-of-the-art models, if possible.

Annotated reviews are not available for download in order to protect the identity of reviewers who chose to remain anonymous.

·

Basic reporting

The general writing style in this paper is legible and understandable but the authors sometimes make common grammar mistakes such as omitting the "s" in first person singular verbs or omitting some articles that should be present.

The Introduction section is well written and describes the motivation and the objective of the study. I believe that the sentence about representing each segmented position by 0 or 1 (lines 59-61) is difficult to understand: where are these characteristics represented and how are they related to each position?

The references along the article are appropriate.

The article follows the common section structure in computer science (and more loosely the PeerJ standard), including a section for related work, sections to explain the proposed solution and a section for results and discussion.

Figures are of good quality in general. I would suggest adjusting the darkness of the green color used in Figures 1 and 4 (make it lighter so that there is a higher contrast between the background and the foreground text) as well as making legends in Figure 4 way larger, as they are currently illegible. Figure 3 should include a description of the numbers in parentheses across the image, are they the dimension of each layer output? The colors applied to the layers should be described as well if they hold some meaning, e.g. are all red layers a dense feed forward operation?

Original data appears to have been collected from a website, although there is no apparent direct download for the datasets, so the authors must have composed the dataset themselves out of the statistics available. The dataset has been published by the authors in a GitHub repository.

Experimental design

This article seems to fit within the scope of journal PeerJ Computer Science, as it applies widely known machine learning methods to a concrete problem and develops improvements on the models and a specific solution to tackle the training and prediction for this kind of data.

The authors aim to tackle a problem of modeling (european) football plays using more modern tools (neural networks) that have not been used yet for this objective.

The fundamental part of the model description is Figure 3 which details the structure of the neural network used, but I think it could be clearer if it could somehow indicate the number of actual dense layers that are used (I believe it could be every red and yellow layer, but it should be clearer if some researcher wants to reproduce this work).

I think that the learning rate used in the Adam optimizer (0.01, Table 6) is much higher than the standard 0.001, is there a reason behind this?

Validity of the findings

The resulting metrics seem reasonable and the stated conclusions are supported by the results.

Since the baseline model performs better than your newly proposed model (when both are without feature engineering), could it improve further than yours when the engineered features are provided?

Additional comments

My initial concern with the approach of the authors was that feature engineering is supposed to be less useful with deep learning models but, in this case, the results show that this step is key in finding a well-performant model. My question now is whether the feature engineering is providing more improvements than the predictor itself and, as a result, if one could achieve even better metrics by providing the engineered metrics to another predictor (e.g. the baseline). The overall quality of the article, however, is good and I would consider it ready for acceptance after fixing the details I indicated above.

---

## Round 0.2 · accepted · Accept

Both reviewers consider the paper is ready for publication... and so do I. Congratulations!

Reviewer 1 ·

Basic reporting

The manuscript is acceptable in its current form.

Experimental design

The authors have responded all the previous queries/revisions.

Validity of the findings

The authors surpassed the baseline models and included a full performance analysis of their proposal.

·

Basic reporting

no comment

Experimental design

no comment

Validity of the findings

no comment

Additional comments

The authors have addressed all of my concerns adequately in the rebuttal. I no longer hold any issues against the publication of the article.